# Evaluation of Plasma Circulating Cell Free DNA Concentration and Integrity in Patients with Prostate Cancer in Jamaica: A Preliminary Study

**DOI:** 10.3390/diseases8030034

**Published:** 2020-09-07

**Authors:** Andrew Condappa, Donovan McGrowder, William Aiken, Wayne McLaughlin, Maxine Gossell-Williams

**Affiliations:** 1Department of Basic Medical Sciences (Pharmacology Section), Faculty of Medical Sciences, The University of the West Indies, Kingston 7, Jamaica; andrewdappa@gmail.com (A.C.); maxine.gossell@uwimona.edu.jm (M.G.-W.); 2Department of Pathology, Faculty of Medical Sciences, The University of the West Indies, Kingston 7, Jamaica; 3Department of Surgery, Radiology, Anaesthesia and Intensive Care, Section of Surgery, Urology Division, Faculty of Medical Sciences, The University of the West Indies, Kingston 7, Jamaica; william.aiken@uwimona.edu.jm; 4Department of Basic Medical Sciences (Molecular Biology Section), Faculty of Medical Sciences, The University of the West Indies, Kingston 7, Jamaica; wayne.mclaughlin@uwimona.edu.jm

**Keywords:** cell free, circulating DNA, prostate, cancer, integrity

## Abstract

Background: Cell free circulating DNA (cfcDNA) is a promising diagnostic tool for prostate cancer (PCa). This study aimed to measure the cfcDNA concentration and integrity in PCa patients using quantitative polymerase chain reaction (qPCR) analysis. This study also assessed the correlation between these molecular biomarkers with total prostate-specific antigen (PSA), Gleason score, prostate volume, and age. Methods: Eleven PCa patients and 9 persons with benign prostatic hyperplasia (BPH) were recruited. Blood samples were collected before prostate biopsy and plasma quantified by qPCR amplification of the ALU 115 DNA sequence, with the ratio of ALU 247 to ALU 115 reflecting cfcDNA integrity. Results: There were no significant differences in median, interquartile range (IQR) cfcDNA concentration or cfcDNA integrity between the patients with PCa (47.9 (214.93) ng/mL; 0.61 (0.49)) and persons with BPH (41.5 (55.13) ng/mL, *p* = 0.382; 0.67 (0.45), *p* = 0.342). A weakly positive correlation exists between cfcDNA concentration and total PSA (r = 0.200, *p* = 0.555) but not with age or Gleason score in PCa patients. Conclusion: cfcDNA concentration was relatively nonsignificantly higher in PCa patients in comparison to persons with BPH, whereas cfcDNA integrity was similar in both groups. Though limited in sample size, this study shows that cfcDNA concentration may be a potentially valuable noninvasive biomarker for the diagnosis of PCa.

## 1. Introduction

In the diagnosis and monitoring of prostate cancer (PCa), the most notable biomarker is prostate-specific antigen (PSA). PSA is detectable in the blood because some of it escapes across the basement membrane, enters the interstitium of the prostate gland, and is absorbed by the lymphatic system and blood circulation [1]. While there is no level of PSA that precludes the possibility of PCa, the normal generic serum PSA range is accepted as 0.0 to 4.0 ng/mL. A PSA level of 4.0 ng/mL or more is indicative of an increased risk for PCa [2]. However, elevated PSA on its own is not specific for the diagnosis and monitoring of patients with PCa. This is primarily due to the fact that PSA levels can increase in other prostatic diseases, such as prostatitis and benign prostatic hyperplasia (BPH) [3].

The PSA test is complemented by the digital rectal exam (DRE) to increase diagnostic accuracy. However, the findings on DRE are also not specific for PCa. The primary means of diagnosing PCa is through histopathologic confirmation of a prostate biopsy. Biopsies also have their own limitations such as being an invasive procedure and increasing the risk of serious infection [2]. Consequently, there is great need for new markers or indicators for PCa diagnosis that are accurate, reliable, and less invasive. One potential biomarker that fits these criteria for PCa is circulating cell free DNA (cfcDNA).

The chief source of cfcDNA in the plasma of healthy persons is apoptotic cells which cleave their own DNA into shorter fragments and secrete homogenous truncated DNA fragments with length of approximately 185–200 base pairs [4]. Conversely, DNA fragments are secreted from malignant cells as a result of molecular processes such as autophagy and catastrophic mitosis, mitochondrial catastrophe, or necrosis with variable length owing to indiscriminate and inadequate digestion of genomic DNA [5]. This results in increased levels of cfcDNA possessing longer fragments in plasma or serum of subjects with cancer [6].

Circulating cell free DNA is a noninvasive and sensitive biomarker, and a number of methods have been employed for its measurement in plasma and serum including quantitative polymerase chain reaction (PCR) using PicoGreen assay [7], real-time quantitative PCR, or immunological methods such as enzyme-linked immunosorbent assay (ELISA) [8] and quantitative, multiplex PCR for circulating nuclear or mitochondrial DNA [9]. Circulating cell free DNA may be used for the early diagnosis of primary breast and pancreatic cancers [10,11] as well as for monitoring of tumor burden following treatment [12], residual disease in patients with specific cancer subtypes after surgery [13], and resistance to treatment [14].

The clinical utility of cfcDNA concentrations and cfcDNA integrity has been investigated in many studies including prostate cancer [15,16,17]. Chun et al. (2006) examined cfcDNA concentrations in 142 men diagnosed with PCa and 19 men with BPH and reported significantly higher values in men with PCa than their counterparts with BPH [18]. Similar findings were reported by Wroclawski et al. (2013) in a study which evaluated the plasma cfcDNA levels in 133 men with PCa and 33 healthy controls that confirmed significantly higher concentrations of cfcDNA in PCa patients [19]. Although both studies reported elevations in cfcDNA concentrations, the findings are limited, as there were no indications of exclusion of confounding factors that have been identified to increase cfcDNA levels such as autoimmune and inflammatory diseases [20].

Umetani et al. (2006) conceptualized a method using real-time quantitative PCR to directly measure cfcDNA integrity [21]. Circulating cell free DNA integrity was calculated as the ratio of repeated sequences of ALU 115 and ALU 247. ALU elements are recognized as the most abundant sequences in the human genome, with a copy number of about 1.4 million per genome. ALU sequences are short interspersed elements which account for more than 10% of the genome [22]. Umetani et al. (2006) designed ALU 115 primers to amplify both short and long DNA fragments representing the total amount of cfcDNA, and ALU 247 primers were designed to amplify only long DNA fragments representing the DNA released from nonapoptotic cells (cancer cells) [21].

This study aimed to measure cfcDNA concentration and integrity in patients with PCa and to compare them to those with BPH. This study aimed at also assessing whether there is a relationship between cfcDNA concentration with baseline PSA, Gleason score, total prostate volume, age, and number of preexisting comorbidities.

## 2. Materials and Methods

The study was reviewed and approved by the ethics committee of The University of the West Indies. The study was a pilot, cross-sectional study that recruited male participants from 1 August 2016 to 31 April 2017 through opportunity (convenience) sampling. Participants were recruited from the urology clinic prior to histological confirmation of prostate cancer with their attending physician. Blood samples were collected from consenting participants who were suspected of having PCa for the measurement of cfcDNA concentrations, cfcDNA integrity, and total prostate-specific antigen (tPSA). Patients with a negative or positive biopsy for prostate cancer were included in this study. Data collection involved chart review to determine patients’ demographic, clinicopathologic, and biochemical information. A correlation of the number of comorbidities present associated with cfcDNA concentration was also investigated.

### 2.1. Testing Blood Samples for Total PSA and Circulating Cell Free DNA Concentration and Integrity

#### 2.1.1. Blood Sample Collection

A blood sample of 8 mL was collected from each consented participant after recruitment to determine cfcDNA concentration, cfcDNA integrity, and tPSA before the initiation of therapeutic intervention.

#### 2.1.2. Measurement of Total PSA

The measurement of serum tPSA was performed using a cobas 6000 analyzer (Roche Diagnostics, Indianapolis, IN, USA) in the Chemical Pathology Laboratory, The University of the West Indies. The Roche Elecsys tPSA assay is centered on a sandwich electrochemiluminescent immunoassay test principle that involves the reaction of a monoclonal PSA-specific antibody labeled with ruthenium complex and a biotinylated monoclonal PSA-specific antibody to form a complex. There is a resulting generation of chemiluminescent emission measured by a photomultiplier and the determination of tPSA concentration by the use of a calibration curve [23].

#### 2.1.3. Preparation of the Plasma Sample for DNA Extraction

Four milliliters of blood were collected in an ethylenediaminetetraacetic acid (EDTA) tube and processed within 2 h. The blood collected from patients was centrifuged at 1200 rpm for 10 min in a Thermo Scientific Cl40 centrifuge (Thermo Fisher Scientific Limited, Massachusetts, MA, USA) to obtain the plasma. The plasma was separated, and a second centrifugation at 2400 rpm for 10 min was performed. Plasma samples were aliquoted into tubes and stored at −20 °C before DNA extraction.

#### 2.1.4. Circulating Cell Free DNA Extraction from Plasma

DNA extraction was performed using the organic DNA extraction method adapted by Xue et al. 2009 [24]. Five hundred microliters of plasma were mixed with 5 μL Triton X-100 (Sigma-Aldrich, Dorset, UK) and heat denatured at 98 °C for 5 min. Samples were placed on ice for 5 min, then extracted with an equal volume of phenol–chloroform–isoamyl alcohol (25:24:1, *v*:*v*:*v*) (Sigma-Aldrich, UK), and centrifuged for 10 min at 14,000× *g*. The aqueous phase was precipitated overnight with 1/10 volume of 3M sodium acetate (NaOAc) and 2.5 volume of 100% ethanol at −20 °C. The DNA pellet was washed with ethanol, air-dried, and resuspended in 50 μL of ddH_2_O.

#### 2.1.5. Circulating Cell Free DNA Quantification

The amount of DNA was determined by a quantitative real-time PCR (qPCR) technique developed by Umetani et al. (2006) [21]. This method used a set of primers to amplify segments of ALU sequences. A set of primers for the 115 base-pair amplicons was designed to amplify both shorter and long DNA fragments representing the total amount of circulating cell free DNA [25]. A second set of primers for the 247-bp amplicons was also designed to amplify only long DNA fragments representing the DNA released from nonapoptotic cells. The sequences of the ALU 115 primers used were forward 5′CCTGAGGTCAGGAGTTCGAG-3′ and reverse 5′CCCGAGTAGCTGGGATTACA-3′. The sequences of the ALU 247 primers used were forward 5′GTGGCTCACGCCTGTAATC-3′ and reverse 5′CAGGCTGGAGTGCAGTGG-3′. CfcDNA integrity was calculated as the ratio of concentrations in each assay:
(1)$$\text{CfcDNA integrity}=\frac{\mathrm{concentration}\;\mathrm{of}\;247\;\mathrm{bp}\;\mathrm{fragments}}{\mathrm{concentration}\;\mathrm{of}\;115\;\mathrm{bp}\;\mathrm{fragments}}$$


The standard reaction volume was 20 μL and consisted of 5 μL of isolated templates of DNA sample, 0.2 μL of forward and reverse primer, and 10 μL of SYRB Green Master Mix (Qiagen, Maryland, MD, USA). The real-time PCR (qPCR) was carried out using The LightCycler^®^ 480 Real-Time PCR System (Roche Diagnostics, Mannheim, Germany) and involves pre-cycling heat activation of DNA polymerase at 95 °C for 10 min, followed by 35 cycles of denaturation at 95 °C for 30 s, annealing at 64 °C for 30 s, and extension at 72 °C for 30 s.

The DNA levels were calculated from a standard curve with serial dilutions (10 ng–0.01 pg) and were prepared with standardized genomic DNA (standardized genomic from a qPCR kit). For each plate, a negative control was used for quality control, and mean values were calculated from triplicate quantification reactions. 

#### 2.1.6. Data Collection

Participants were interviewed and their demographic, clinical, and pathological data (collected from dockets); results from qPCR analysis; and tPSA results were entered on a data collection form.

#### 2.1.7. Data Analysis

Demographic data were evaluated using descriptive statistics. Continuous variables were expressed as medians, interquartile range (IQR), and categorical variables were expressed as counts and percentages. The relationship between cfcDNA concentration and integrity with continuous variables, such as age, was investigated using Mann–Whitney U test. The relationship between cfcDNA concentration and integrity with categorical variables was investigated using the Chi-square test or, where appropriate, Fisher’s exact test. The association between cfcDNA concentration and cfcDNA integrity with baseline tPSA, Gleason score, total prostate volume, age, and the presence and number of comorbidities was assessed using Spearman’s correlation, with two-sided *p* values less than 0.05 considered statistically significant. The statistical analysis was done using Statistical Package for Social Sciences, version 20 (SPSS, Inc., Chicago, IL, USA).

## 3. Results

Thirty-five (35) eligible patients were approached, and 29 patients consented to participating in the study. Consenting patients were recruited at a rate of approximately one patient per week. Twenty (20/29, 69%) of these patients completed their biopsy for histological confirmation for prostate cancer (PCa); the remaining patients’ results (9/29, 31%) were not available (NA). Eleven (11/20, 55%) patients were confirmed to have PCa, and 9 (45%) patients had BPH. Table 1 provides the characteristic of the participants.

The median (IQR) cfcDNA concentrations for PCa and BPH patients were 47.9 (214.93) ng/mL and 41.5 (55.13) ng/mL, respectively (Figure 1). There was no statistical difference between the groups (Mann–Whitney U test, *p* = 0.382). The median (IQR) cfcDNA integrity for PCa and BPH patients were 0.6159 (0.49) and 0.6734 (0.45), respectively (Figure 2). There was also no statistical difference between the two groups (Mann–Whitney U test, *p* = 0.342).

There was a weakly positive correlation between cfcDNA concentration and serum total PSA (r = 0.200, *p* = 0.555) but not with other clinicopathological parameters such as age and Gleason score in PCa patients (Table 2). Likewise, there was also no significant correlation between cfcDNA integrity and age, serum tPSA, total prostate volume, and number of comorbidities in PCa patients (Table 2).

## 4. Discussion

Circulating cell free DNA is a promising and interesting biomarker for PCa. This pilot study attempted to examine the distinguishing capabilities of plasma cfcDNA concentration and cfcDNA integrity for PCa and BPH as well as to assess the impact of confounding variables. The median cfcDNA concentration and cfcDNA integrity were assessed and calculated by the use of qPCR that applied the two genomic markers (ALU 155 and ALU 247) previously developed by Umetani et al. (2006) [21].

To the best of our understanding, this is the first investigation of the importance of cfcDNA concentration and cfcDNA integrity using ALU 115 and ALU 247 sequences in plasma from Caribbean men of African Ancestry with prostate cancer and the control group composed of BPH patients. In this study, the median cfcDNA concentration for PCa patients was higher in comparison to BPH patients; however, this trend was not statistically significant. Feng et al. (2013) conducted a similar study using ALU markers involving 96 PCa and 112 BPH patients and found significantly higher levels of cfcDNA concentration among PCa patients [17]. Furthermore, other studies reported that PCa patients typically have higher cfcDNA concentrations in comparison to benign prostatic conditions despite variation in analytical approaches [17,18,19,26,27,28,29,30].

There are other studies that have examined the importance of cfcDNA concentration and cfcDNA integrity using the ALU 115-qPCR marker in plasma [31,32,33]. Fawzy et al. using ALU sequence found very significant levels of cfcDNA in PCa patients compared to those with BPH [31]. Similarly, in a recent study, cfcDNA was also significantly higher in the group of patients with PCa compared to BPH group [32]. Another recent study supported these observations, where cfcDNA determined using ALU species was elevated in PCa patients compared to healthy controls [33].

Regarding cfcDNA integrity, this parameter was lower in PCa patients in comparison to BPH persons, but there was no statistical difference between groups. This trend was not consistent with previous studies [17,34]. In particular, Feng et al. (2013) also identified that PCa patients had significantly higher levels of cfcDNA integrity in comparison to BPH patients [17].

The cfcDNA integrity was mildly elevated in BPH persons compared to PCa patients, and these observations diverge from the pattern reported by other investigators for both cfcDNA ALU 115 and ALU 247 sequences [31,32,33]. In two of these studies, very significant cfcDNA integrity levels were observed in PCa patients compared to BPH [31,32]. In another study, similar observations of elevated cfcDNA integrity were found in PCa patients compared with their healthy control counterparts [33]. The mildly decreased cfcDNA integrity in PCa patients in our study may be attributed to lower ALU 247 fragments. There are studies that have shown that the ALU 247 sequence is higher in metastatic PCa compared to nonmetastatic PCa patients [31,32]. This could significantly impact the level of significance of cfcDNA integrity in different groups of PCa patients in one study compared with another.

Prostate cancer is believed to progress via a stepwise development that involves the conversion of benign prostatic epithelial cells to high-grade prostatic intraepithelial neoplasia and eventually to invasive adenocarcinoma [35]. The transitional process comprises a number of somatic molecular modifications that take place before or at the onset of high-grade prostatic intraepithelial neoplasia. These genetic alterations include hypermethylation of the Glutathione S-transferase P1 (GSTP1) promoter, the stimulation of the proto-oncogene Myc, and deletions of regions protecting recognized tumor suppressors on chromosome 10q23 such as phosphatase and tensin homolog (PTEN) [36,37]. Likewise, expression of PTEN was downregulated in PCa patients [38], and a later study reported that decreased PTEN expression was 30 times more probable in PCa cases compared with subjects with BPH [39]. Furthermore, the c-Myc proto-oncogene has a critical role in cell propagation, differentiation, and programmed death [40]. Significantly elevated c-Myc expression was found in PCa patients [41], while in an earlier study, patients with PCa had 2-fold higher levels of c-myc transcripts than those with BPH [42].

Hypermethylation of the GSTP1 promoter leads to its expression in PCa patients compared with those with BPH, signifying its critical role in prostate cancer progression [43,44]. Hypermethylation of the GSTP1 gene has a sensitivity ranging from 11 to 100%, with specificity ranging from 93 to 100%. Additionally, the inclusion of additional markers increased the sensitivity and specificity for PCa diagnosis [29]. Moreover, cfcDNA remains a reliable source to evaluate molecular biomarkers of PCa, and microsatellite alterations is another molecular biomarker that has been investigated [29,45]. The detection of microsatellite instability from cfcDNA has a sensitivity ranging from 34 to 57%, specificity ranging from 70 to 100% [29].

In this study, there was a weakly positive correlation between cfcDNA concentration and total PSA but not with other clinicopathological parameters such as age and Gleason score in PCa patients. Feng et al. showed a significant association between cfcDNA concentration and total PSA. Circulating cell free DNA concentration and cfcDNA integrity could distinguish prostate cancer from BPH in patients with serum total PSA ≥ 4 ng/mL [17]. Furthermore, there was an observed much stronger inverse correlation between cfcDNA integrity and serum total PSA (r = −0.455, *p* = 0.160). There are disparities in a number of studies examining the relationship between cfcDNA concentration, cfcDNA integrity, and total PSA. Khani et al. demonstrated a significant direct association between cfcDNA concentration and low total PSA in prostate cancer patients, as approximately 20% with normal PSA levels had higher cfcDNA concentrations compared to those persons with higher total PSA values [32]. In another study, Fawzy et al. reported no statistically significant association between cfcDNA integrity and total PSA level [31], which was in agreement with findings of other studies [19,46,47].

In this study, the lack of statistically significant difference with cfcDNA concentration as well as cfcDNA integrity in the two groups of patients may be associated with the small sample size. Furthermore, cfcDNA concentration and cfcDNA integrity may vary due to sample preparation. Serum typically has higher cfcDNA concentrations in comparison to plasma, and this is primarily due to the coagulation of serum and subsequent leukocyte lysis, which releases additional DNA [48]. Extraction techniques may also contribute to disparities in cfcDNA concentration [49].

The determination of cfcDNA concentrations present some limitations such as variations in methodology as well as the absence of standardization in existing methodologies which impede the application of these biomarkers in clinical practice [50,51]. Moreover, the diagnostic potential of cfcDNA concentration and cfcDNA integrity have been found to be elevated not only in prostate cancer [17] but also in other cancers, such as rectal [52], colorectal [53], hepatocellular [54], breast [55], and periampullary [56]. Thus, its utility in the clinical setting is limited by a lack of disease specificity.

## 5. Conclusions

The current study found that cfcDNA (ALU 115 sequence) levels were relatively nonsignificantly higher in PCa patients in comparison to BPH persons, whereas cfcDNA integrity was somewhat similar in both groups. There was a weakly positive correlation between cfcDNA and tPSA but not with other clinicopathological parameters such as age and Gleason score in PCa patients.

The most significant limitation of this study is the sample size, which may have impacted the concentration of cfcDNA, the cfcDNA integrity in PCa patients, and the significance level. Therefore, although the cfcDNA concentration in plasma is nonsignificantly higher in PCa patients, it may be a potentially valuable noninvasive biomarker for screening and diagnosis of the disease. Our study is preliminary and requires a larger sample size for further investigations to confirm this assumption.

## Figures and Tables

**Figure 1 diseases-08-00034-f001:**
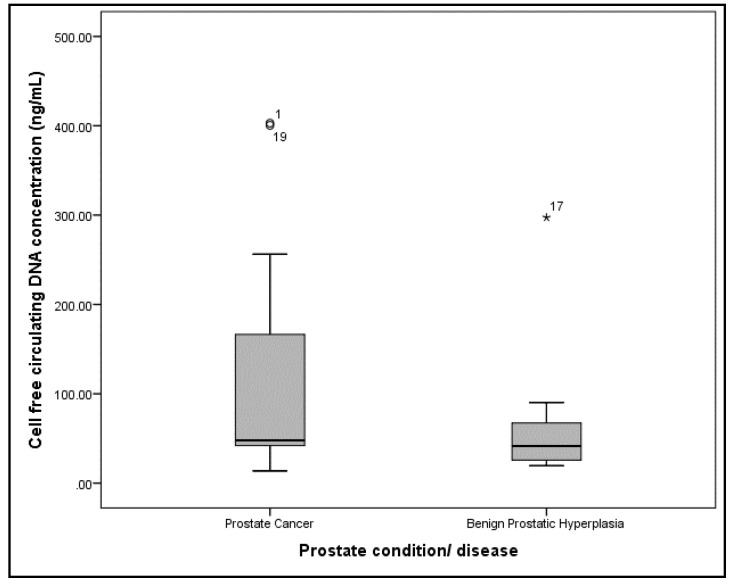
Box plot for the plasma cell free circulating DNA concentration of prostate cancer and benign prostatic hyperplasia patients. Note that ◦ and * represents outliers.

**Figure 2 diseases-08-00034-f002:**
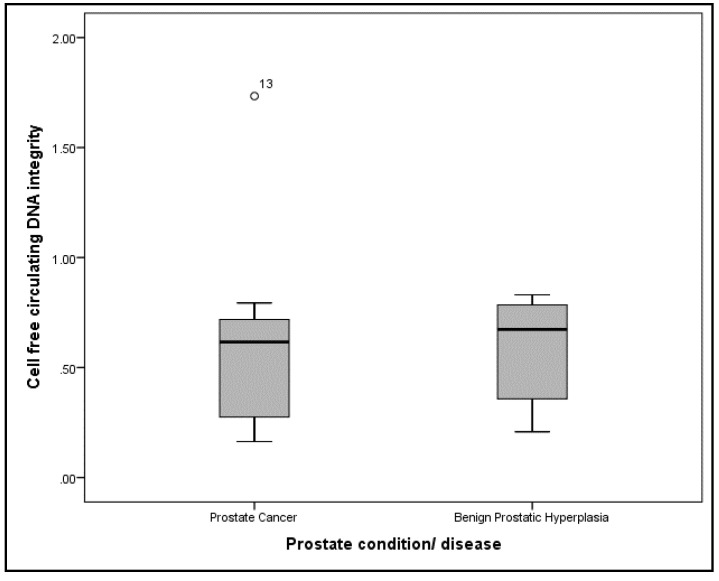
Box plot for the of plasma cell free circulating DNA integrity of prostate cancer and benign prostatic hyperplasia patients. Note that ◦ represents an outliers.

**Table 1 diseases-08-00034-t001:** Patient characteristics.

Characteristics	PCa (*n* = 11)	BPH (*n* = 9)	*p*
Age in years, median (IQR)	68 (7)	65 (8)	0.236
Type of comorbidities, *n* (%)			
Hypertension	5 (45.5%)	2 (22.2%)	-
Diabetes	3 (27.3%)	2 (22.2%)	-
Renal Calculi	1 (9.1%)	2 (22.2%)	-
Other	2 (18.2%)	1 (11.1%)	-
Serum total PSA in ng/mL, median (IQR)	24.2 (152.4)	8.2 (5.05)	0.138
Serum total PSA range, *n* (%)			
≥4 ng/mL	11 (100%)	8 (88.9%)	0.450
<4 ng/mL	0 (0.0%)	1 (11.1%)	
Prostate volume in cc, median (IQR)	54 (28.3) *a	113.8 (125.1) *b	0.011
Gleason Score, median (IQR)	7 (3)	-	-
Gleason score, *n* (%)			
<7	4 (36.4%)	-	-
=7	3 (27.3%)	-	-
>7	4 (36.4%)	-	-

*a, value representative for 7 patients with PCa. *b, value representative for 8 patients with BPH.

**Table 2 diseases-08-00034-t002:** Correlation between cell free circulating DNA and cell free circulating DNA integrity different variables of patients with prostate cancer.

Variables	*n*	Cell Free Circulating DNA Concentration		Cell Free Circulating DNA Integrity	
		r_s_	*p*	r_s_	*p*
Age	11	−0.110	0.747	−0.303	0.395
Serum total PSA	11	0.200	0.555	−0.455	0.160
Gleason score	11	−0.119	0.728	−0.052	0.879
Total prostate volume	11	−0.179	0.702	0.071	0.879
Number of comorbidities	11	−0.162	0.633	0.249	0.461

rs, Spearman’s correlation coefficient, NA, not applicable.

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
