# Peer review of "Evaluation of Plasma Circulating Cell Free DNA Concentration and Integrity in Patients with Prostate Cancer in Jamaica: A Preliminary Study"

_diseases, 2020, doi:10.3390/diseases8030034_

Round 1

Reviewer 1 Report

The manuscript by Condappa describes the evaluation of Plasma Cell-Free Circulating DNA Integrity in Patients with Prostate Cancer in Jamaica.

The study gave an excellent rationale to consider the cfcDNA for diagnosis of prostate cancer since the current diagnostic strategy has some limitations. Especially at the time of overdiagnosis by PSA and precision medicine due to prostate cancer heterogeneity, the present study is exciting. However the small sample size and only the amount of cfcDNA makes this study of less significance in the present form. The information obtained from this article is of revealed by many earlier studies. Generally the level of cfcDNA is lower in prostate cancer due to the tumor localization and its distant access to the circulatory system. In this conditions the authors should clearly describe how the current study may forma a foundation to change the current diagnosis methods.  There are various genetic events reported to occur or drive the prostate cancer progression such as pten, c-myc, GST PAP etc.  Authors should check for any change in genetic changes in the cfcDNA of tumor versus BPH samples.

Author Response

The manuscript by Condappa describes the evaluation of Plasma Cell-Free Circulating DNA Integrity in Patients with Prostate Cancer in Jamaica.

The study gave an excellent rationale to consider the cfcDNA for diagnosis of prostate cancer since the current diagnostic strategy has some limitations. Especially at the time of overdiagnosis by PSA and precision medicine due to prostate cancer heterogeneity, the present study is exciting.

  • However the small sample size and only the amount of cfcDNA makes this study of less significance in the present form.

  • See Discussion and conclusion (Pages 7).

  • In this study the lack of statistically significant difference with cfcDNA concentration, as well as cfcDNA integrity in the two groups of patients may be associated with the small sample size.

  • The most significant limitation of the study is the sample size which may have impacted the concentration of cfcDNA and cfcDNA integrity in the PCa patients, and the significance level. Therefore, although the cfcDNA concentration in plasma is non-significantly higher in PCa patients it may be a potentially valuable non-invasive biomarker for screening and diagnosis of the disease. Our study is preliminary and requires a larger sample size for further investigations to confirm this assumption.

The information obtained from this article is of revealed by many earlier studies.

Generally the level of cfcDNA is lower in prostate cancer due to the tumor localization and its distant access to the circulatory system.

  • In this conditions the authors should clearly describe how the current study may forma a foundation to change the current diagnosis methods.

  • Based on the finding the author indicated on page 7 that:
  • Therefore, although the cfcDNA concentration in plasma is non-significantly higher in PCa patients it may be a potentially valuable non-invasive biomarker for screening and diagnosis of the disease. Our study is preliminary and requires a larger sample size for further investigations to confirm this assumption.

  • There are various genetic events reported to occur or drive the prostate cancer progression such as pten, c-myc, GST PAP etc. Authors should check for any change in genetic changes in the cfcDNA of tumor versus BPH samples.

  • Page 6. Prostate cancer is believed to be progress via a stepwise development that involves the conversion of benign prostatic epithelial cells to high-grade prostatic intraepithelial neoplasia and eventually to invasive adenocarcinoma…

Kindly note that changes were made to the title of the manuscript which now reads:

Evaluation of Plasma Circulating Cell Free DNA Concentration and Integrity in Patients with Prostate Cancer in Jamaica

The manuscript was also reviewed for grammatical and typographical errors.

Reviewer 2 Report

Condappa et al. investigated the concentration and integrity of cell free circulating DNA from plasma of patients with prostate cancer or BPH. However, they could not detect significant differences. Furthermore, there were no correlations with clinico-pathological parameters such as age, PSA, Gleason Score etc.

The greatest limitation of this study is the small sample size (11 prostate cancer and 9 BPH). Furthermore, there are some studies that should be discussed:

  1. Circulating cell-free DNA integrity as a diagnostic and prognostic marker for breast and prostate cancers
  2. The value of the plasma circulating cell-free DNA concentration and integrity index as a clinical tool for prostate cancer diagnosis: a prospective case-control cohort study in an Iranian population
  3. Quantitative analysis of plasma cell-free DNA and its DNA integrity in patients with metastatic prostate cancer using ALU sequence

Table 1: Why are the median numbers of comorbidities 1 and 0 for prostate cancer and BPH patients, respectively, when more patients havecomorbidities?

Methods: Which qPCR machine was used?

Minor:

In the abstract, the unit ng/ml for DNA integrity needs to be omitted.

Abbreviations once introduced (e.g., PCa, cfcDNA, BPH, qPCR) need to be used throughout the manuscript.

Table 1: TNM stage is not presented but T stage. Since T stage is only available for one patient, it should not be reported that the relationship between concentration and integrity of cell free circulating DNA and T stage was evaluated.

Author Response

Condappa et al. investigated the concentration and integrity of cell free circulating DNA from plasma of patients with prostate cancer or BPH. However, they could not detect significant differences. Furthermore, there were no correlations with clinico-pathological parameters such as age, PSA, Gleason Score etc.

The greatest limitation of this study is the small sample size (11 prostate cancer and 9 BPH).

  • Furthermore, there are some studies that should be discussed:

Circulating cell-free DNA integrity as a diagnostic and prognostic marker for breast and prostate cancers (study 1).

The value of the plasma circulating cell-free DNA concentration and integrity index as a clinical tool for prostate cancer diagnosis: a prospective case-control cohort study in an Iranian population (study 2).

Quantitative analysis of plasma cell-free DNA and its DNA integrity in patients with metastatic prostate cancer using ALU sequence (study 3).

  • These three studies were discussed (references 31 - 33):
  • Page 6. There are other studies that have examined the importance of cfcDNA concentration and cfcDNA integrity using the ALU115-qPCR marker in plasma [31-33]…
  • The cfcDNA integrity were mildly elevated in BPH persons compared to PCa patients, and these observation diverges from the pattern reported by other investigators who determined both cfcDNA ALU115 and ALU247 sequences [31-33]…

  • Table 1: Why are the median numbers of comorbidities 1 and 0 for prostate cancer and BPH patients, respectively, when more patients have comorbidities?

  • The median (IQR) of the number of comorbidities was deleted. Table 2 (page 12) provides the number of comorbidities in the two groups of patients.

  • Methods: Which qPCR machine was used?

  • Page 4. The real-time PCR (qPCR) was carried out using The LightCycler® 480 Real-Time PCR System (Roche Diagnostics)

Minor:

  • In the abstract, the unit ng/ml for DNA integrity needs to be omitted.

  • Page 1, The unit ng/mL for DNA integrity was deleted.

  • Abbreviations once introduced (e.g., PCa, cfcDNA, BPH, qPCR) need to be used throughout the manuscript.

  • Abbreviations such as PCa, cfcDNA, BPH, qPCR etc. once introduced were consistently used throughout the manuscript.

  • Table 1: TNM stage is not presented but T stage. Since T stage is only available for one patient, it should not be reported that the relationship between concentration and integrity of cell free circulating DNA and T stage was evaluate

  • TNM stage or the TMN stages was deleted/removed from table 1 (page 12) and in the text of the manuscript.

Kindly note that changes were made to the title of the manuscript which now reads:

Evaluation of Plasma Circulating Cell Free DNA Concentration and Integrity in Patients with Prostate Cancer in Jamaica

The manuscript was also reviewed for grammatical and typographical errors.

Round 2

Reviewer 1 Report

I do not have further comments 

Author Response

Thank you

Reviewer 2 Report

The authors now highlight: "There was a weakly positive correlation between cfcDNA concentration and serum total PSA (r = 0.200, p = 0.555) but not with other clinicopathological parameters such as age and Gleason score in
PCa patients (Table 2)." However, there might be a much stronger inverse correlation between cfcDNA integrity and serum total PSA (r = -0.455, p = 0.160).

In the discussion part the authors should check the following sentence: "There are studies that have shown that ALU 247 sequence is higher in metastatic PCa compared to metastatic PCa patients [31, 32]." Metastatic compared to metastatic PCa patients makes no sense.

Author Response

  • The authors now highlight: "There was a weakly positive correlation between cfcDNA concentration and serum total PSA (r = 0.200, p = 0.555) but not with other clinicopathological parameters such as age and Gleason score in 
    PCa patients (Table 2)."

However, there might be a much stronger inverse correlation between cfcDNA integrity and serum total PSA (r = -0.455, p = 0.160). It was expected that prostate cancer patients with higher total PSA levels should have higher levels of ALU 247 and thus higher cfcDNA integrity.

  • In this study there was a weakly positive correlation between cfcDNA concentration and total PSA but not with other clinicopathological parameters such as age and Gleason score in PCa patients. Feng et al. showed a significant association between cfcDNA concentration and total PSA. Circulating cell free DNA concentration and cfcDNA integrity could distinguish prostate cancer from BPH in patients with serum total PSA ≥ 4 ng/mL [17]. Furthermore, there was an observed much stronger inverse correlation between cfcDNA integrity and serum total PSA (r = -0.455, p = 0.160). There are disparities in a number of studies examining the relationship between cfcDNA concentration, cfcDNA integrity, and total PSA. Khani et al. demonstrated significant direct association between cfcDNA concentration and low total PSA in prostate cancer patients, as approximately 20% with normal PSA levels had higher cfcDNA concentrations compared to those persons with higher total PSA values [32]. In an another study, Fawzy et al. reported no statistically significant association between cfcDNA integrity and total PSA level [31] which was in agreement with findings of other studies [46-48].
  • In the discussion part the authors should check the following sentence: "There are studies that have shown that ALU 247 sequence is higher in metastatic PCa compared to metastatic PCa patients [31, 32]." Metastatic compared to metastatic PCa patients makes no sense.

  • The correction is:

There are studies that have shown that ALU 247 sequence is higher in metastatic PCa compared to non-metastatic PCa patients [31, 32].

  • The authors sincerely apologize for the error.